# Moderate to severe acute pain disturbs motor cortex intracortical inhibition and facilitation in orthopedic trauma patients: A TMS study

**Marianne Jodoin**[1,2], **Dominique M. Rouleau**[1,3], **Audrey Bellemare**[1,2], **Catherine Provost**[1], **Camille Larson-Dupuis**[1,2], **Émilie Sandman**[1,3], **Georges-Yves Laflamme**[1,3], **Benoit Benoit**[1,3], **Stéphane Leduc**[1,3], **Martine Levesque**[1,4], **Nadia Gosselin**[1,2], **Louis De Beaumont**[1,3]*

1 Hôpital Sacré-Cœur de Montréal (HSCM), Montreal, QC, Canada, 2 Département de psychologie, de l'Université de Montréal, Montreal, QC, Canada, 3 Département de chirurgie, de l'Université de Montréal, Montreal, QC, Canada, 4 Hôpital Fleury, Montreal, QC, Canada

* louis.de.beaumont@umontreal.ca

**Data Availability Statement:** All relevant data are within the manuscript and its supporting information files.

## Abstract

### Objective

Primary motor (M1) cortical excitability alterations are involved in the development and maintenance of chronic pain. Less is known about M1-cortical excitability implications in the acute phase of an orthopedic trauma. This study aims to assess acute M1-cortical excitability in patients with an isolated upper limb fracture (IULF) in relation to pain intensity.

### Methods

Eighty-four (56 IULF patients <14 days post-trauma and 28 healthy controls). IULF patients were divided into two subgroups according to pain intensity (mild versus moderate to severe pain). A single transcranial magnetic stimulation (TMS) session was performed over M1 to compare groups on resting motor threshold (rMT), short-intracortical inhibition (SICI), intra-cortical facilitation (ICF), and long-interval cortical inhibition (LICI).

### Results

Reduced SICI and ICF were found in IULF patients with moderate to severe pain, whereas mild pain was not associated with M1 alterations. Age, sex, and time since the accident had no influence on TMS measures.

### Discussion

These findings show altered M1 in the context of acute moderate to severe pain, suggesting early signs of altered GABAergic inhibitory and glutamatergic facilitatory activities.

**Funding:** LDB received funding from the Fonds de Recherche du Québec en Santé for this work Grant number: 35117 Website: http://www.frqs.gouv.qc.ca The funders had no role in study design, data collection and analysis, decision to publish, or preparation of the manuscript

**Competing interests:** The authors have declared that no competing interests exist.

## Introduction

Orthopedic trauma (OT) patients are routinely afflicted by pain and it is considered the most common and debilitating symptom reported among this population [1, 2]. Optimal pain control is an OT care priority as pain interferes with trauma recovery and affects outcome [3, 4].

A growing body of research is currently focused on developing alternative pain management techniques to tackle the alarming drawbacks associated with current standards of care. Among these alternatives, transcranial magnetic stimulation (TMS) has gained attention in recent years for its dual role: 1) its ability to objectively assess pain mechanisms; and 2) its potential applicability in pain management. In chronic pain studies, the primary motor cortex (M1) commonly serves as the targeted brain region due to its connections with the nociceptive system and the known effect of pain on motor function [5, 6]. Despite some variability across TMS studies, there is extensive evidence of an altered balance between inhibitory and facilitatory circuits of M1 in various chronic pain conditions (i.e. fibromyalgia, neuropathic pain, complex regional pain syndrome, phantom limb pain, chronic orofacial pain) [7, 8]. These results highlight maladaptive plasticity within the motor system. M1-cortical excitability alterations have been associated with the severity of the clinical symptoms such as pain intensity, hyperalgesia, and allodynia [9, 10], pointing to the value of TMS as an objective tool that reflects functional alterations. Moreover, cortical excitability restoration through repetitive TMS (rTMS), a technique known to induce lasting modulation effects on brain activity through a multiple day session paradigm, has shown some efficacy in reducing the magnitude of pain, even in refractory chronic pain patients [11–16]. Overall, these results support the role of cortical excitability on pain intensity in chronic pain patients and the potential clinical utility of TMS in pain management among this population.

On the other hand, acute pain initiated by an OT, such as following a fracture, has received little to no attention, despite being highly prevalent. With 15% to 20% of all physician visits intended to address pain-related issues [17, 18], management of acute pain following OT still remains medically challenging [19–22]. Knowing that acute and chronic pain belong to the same continuum and that there is clear evidence of success in the use of rTMS in treating chronic pain, this technique could serve as a potential treatment tool in the early phase of fracture pain by tackling key elements of pain chronification. First, however, a better understanding of the involvement of M1-cortical excitability in acute pain is necessary.

From a physiological point of view, it remains unclear whether motor cortical excitability impairments are expected in a context of acute pain following an OT. On one hand, neuroimaging studies suggest that possible disturbances within M1 only arise once chronic pain has developed, with acute and chronic pain exhibiting distinct and non-overlapping brain activation patterns [23–27]. On the other hand, there is evidence supporting alterations of M1-cortical excitability during acute pain states. Indeed, Voscopoulos and Lema highlight early neuroplasticity involvement of GABA inhibitory interneurons following a peripheral insult, which may contribute to later transition to chronic pain [28]. In parallel, Pelletier and colleagues [29] suggested that pain intensity may act as the driving factor leading to M1-cortical excitability alterations rather than the state of chronic pain itself. This assumption was made by authors after obtaining similar M1 deficiency patterns across chronic pain conditions of various origins. Other TMS studies also showed that pain of moderate to severe intensity (score ≥4 on numerical rating scale (NRS)) leads to greater motor cortex impairments [10]. The relationship between pain intensity in the acute state and its impact on cortical excitability parameters appears a relevant target of investigation.

So far, very few studies have looked into the association between acute pain and M1-cortical excitability. These studies have mainly focused on experimental pain models in healthy

subjects. More specifically, acute experimental pain of low-to-moderate intensity induces a generalized state of M1 inhibition, reflecting changes in both cortical and spinal motoneuronal excitability in healthy participants [30–35]. Findings suggest that acute experimental pain can modify cortical excitability of M1, but the result patterns obtained are different from chronic pain states. In parallel, rTMS studies have been shown effective in both alleviating acute experimental pain and modulating alterations in M1-cortical excitability [36, 37]. Taken together, these findings show that M1 alterations can occur in the context of acute pain and that rTMS over M1 can successfully modulate nociceptive afferent information and restore M1 alterations, even for transient pain sensation in healthy controls. However, due to the subjective nature of pain sensation along with intrinsic differences in pain characteristics across conditions and individuals, translation between experimental pain model and clinical pain following an OT is limited. Therefore, if we are to consider the potential clinical utility of rTMS in alleviating acute pain, studies need to be conducted in a clinical population.

This study therefore aims to assess acute M1-cortical excitability functioning through well-established TMS paradigms according to pain intensity in patients who are in the acute pain phase following an isolated upper limb fracture (IULF). We hypothesize that M1-cortical excitability alterations will be found in patients with higher levels of pain compared to healthy controls and to IULF patients with mild pain.

## Materials and methods

This work was approved by the Hôpital du Sacré-Coeur de Montréal' Ethics Committee (Approval number: 2017–1328). A written consent was obtained by all participating subjects prior to the start of the study. A financial compensation was given to all subjects for their participation.

### Participants

Our sample included 1) patients who have suffered from an isolated upper limb fracture (IULF) and 2) healthy controls. Patients with an IULF were initially recruited from various orthopedic clinics affiliated to a Level 1 Trauma Hospital. To be included in the study, patients had to be aged between 18 and 60 years old and have sustained an IULF (one fractured bone from upper body extremities) within 14 days post-injury. Recruitment of IULF patients took place on the day of the first medical appointment at the orthopedic trauma clinic with the orthopedic surgeon. Testing was conducted within 24 hours post-medical consultation. All testing measures had to be completed prior to surgical procedures (if any) given the known impact of surgery on increased inflammatory response and pain perception [38]. Exclusion criteria consisted of a history of traumatic brain injuries, a diagnosis of and/or a treatment for a psychiatric condition in the last ten years, musculoskeletal deficits, neurological conditions (i.e. epilepsy), chronic conditions (cancer, uncontrolled diabetes, cardiovascular illness, high blood pressure), the use of central nervous system-active medication (hypnotics, antipsychotics, antidepressant, acetylcholinesterase inhibitor, anticonvulsant), history of alcohol and/or substance abuse, acute medical complications (concomitant traumatic brain injury, neurological damage, etc.), and being intoxicated at the time of the accident and/or at the emergency visit. Of note, IULF patients were not restrained from using analgesic medication (acetaminophen, ibuprofen, opioids, etc.) during testing to assure comfort and to avoid interfering with pain management.

The control group consisted of healthy right-handed adults recruited through various social media platforms. As per usual practice in conducting M1 TMS studies, only right-handed control participants were selected as stimulation over non-dominant M1 has been associated with

accentuated within-subject variability [39, 40]. They self-reported to be free of all previously mentioned exclusion criteria.

Study participants were also screened for TMS tolerability and safety [41].

## Assessment measures

Total assessment procedures (including consent) were conducted over a single, 90-minute session. First, participants were invited to complete self-administered questionnaires to gather demographic information and clinical outcome measures (pain intensity and functional disability indices). More specifically, demographic data such as age, sex, and level of education were documented and used to ensure homogeneity between groups.

**Clinical outcome: Pain intensity and functional disability indices.** To assess the perceived level of pain at the time of testing, the numerical rating scale (NRS), a routinely used standardized generic unidimensional clinical pain questionnaire, was administered [42, 43]. To complete the NRS, participants had to circle a number that best fit their current level of pain on the 11-point pain intensity scale, with numbers ranging from 0 ("no pain") to 10 ("worst possible pain"). In order to test the hypothesized impact of acute pain intensity on M1 cortical excitability, IULF patients were divided into two distinct groups according to NRS score: 1) IULF patients who self-reported moderate to severe pain intensity (NRS $\geq 4$ out of 10); 2) IULF patients with mild pain intensity (NRS $< 4$). The cut-off pain intensity scores are based on previous pain studies [10, 44, 45].

The disabilities of the Arm, Shoulder, and Hand (DASH) questionnaire was used as a tool to assess an individual's ability to perform common specific everyday activities relying on upper extremity limbs [46, 47]. This questionnaire consists of 30 items, including 6 that are symptom-related and 24 that are function-related, where patients were asked to rate the level of disability on each activity as experienced since their accident. Continuum of scores on this questionnaire varies between 0 (no disability) and 100 (extreme difficulty).

**Comprehensive assessment of M1 cortical excitability using TMS.** To assess M1 cortical excitability, a TMS figure-of-eight stimulation coil (80mm wing diameter), attached to a Bistim$^2$ Magstim transcranial magnetic stimulators (*Magstim Company*, Whitland, Dyfed, UK), was used. The TMS-coil was positioned flat on the scalp over M1 at a 45˚ angle from the midsagittal line, with its handle pointing backwards. In the IULF group, the TMS coil was positioned over M1 contralaterally to the injury, whereas in the control group, the TMS-coil was systematically positioned over the dominant left hemisphere. Motor evoked potentials (MEP) recordings from the abductor pollicis brevis (APB) was performed using three electrodes positioned over the belly of the target muscle (active electrode (+)), between the distal and proximal interphalangeal joints of the index (reference (-)), and on the forearm (ground). Optimal stimulation site was determined based on the coil position which evoked highest peak-to-peak MEP amplitudes from the target muscle. We used a 3D tracking system (Northern Digital Instruments, Waterloo, Canada) to ensure accurate and consistent TMS coil positioning on the targeted site.

Various well-established TMS protocols were conducted to investigate M1 excitatory and inhibitory mechanisms using single and paired-pulse paradigms. Single pulse magnetic stimulations were first used to establish the resting motor threshold (rMT), i.e. the minimal stimulation intensity needed to elicit a MEP of at least 0.05mV in five out of ten trials [48]. An interstimulus interval, varying from 8 to 10 seconds, was applied to control for possible residual effects of TMS stimulation on M1 activity [49]. The sequence of stimulation intensity was randomly generated by a computer. Short intra-cortical-inhibition (SICI) and facilitation (ICF) were measured via a classic paired-pulse paradigm [50, 51]. The latter protocol involves

the application of two successive TMS pulses, the first pulse set at 80% of the rMT intensity (subthreshold; conditioning stimulus) and the second pulse set at 120% of the rMT (suprathreshold; test stimulus) separated by an interstimulus interval (ISI) of a predetermined duration [50]. To test for SICI, a measure attributed to $GABA_A$ interneurons and receptors activity [52], one sequence of 10 paired-pulse stimulations was completed with an ISI set at 3ms. To test for ICF, one sequence of 10 stimulations was performed with ISI set at 12ms. Measure of ICF is thought to be mediated by excitatory glutamatergic interneurons and N-methyl-D-aspartate (NMDA) receptors [52–56]. Results of SICI and ICF are expressed as percentage ratios of MEP amplitudes. These ratios represent the mean MEP amplitude of paired TMS over the mean MEP amplitude of the test stimuli baseline measurement (10 single magnetic pulses set at 120% rMT). Therefore, high SICI values reflect a lack of intracortical inhibition, whereas a low value ICF corresponds to a lack of intracortical facilitation. Finally, we measured long-interval cortical inhibition (LICI) through paired-pulse TMS of identical suprathreshold intensity (i.e. 120% rMT) with an ISI of 100ms. The first pulse corresponded to the conditioning stimulus whereas the second pulse was the test stimulus. LICI is primarily known to be mediated by $GABA_B$ receptors [57, 58]. To calculate LICI, we used the percentage ratio between the mean peak-to-peak MEP amplitude of the test stimulus response (TSR) and the mean peak-to-peak MEP amplitude of the conditioning stimulus response (CSR) expressed as: mean (TSR)/mean(CSR).

## Statistics

Statistical analyses were performed using IBM SPSS Statistics software version 25 (Armonk, NY, United States). The Shapiro-Wilks test was used to determine the normality of the data. Parametric and nonparametric tests were performed, where appropriate, with a α-level fixed at 0.05. Descriptive analyses were used to characterize and compare the three groups (1- IULF patients with NRS≥4; 2- IULF patients with NRS<4; 3- healthy controls) in our study sample. Results from descriptive analyses are expressed as means, standard deviation (SD), and percentages. We used a Student's t-test or a Mann-Whitney U test to investigate group differences on TMS measures. An analysis of variance (ANOVA) or the Kruskal-Wallis test were also used where appropriate. Pearson and Spearman's correlation analysis were also computed to assess the relationship between functional disability outcomes and the other outcome measures of interest (pain intensity and TMS measures). We corrected for multiple comparisons using False Discovery Rate (FDR) where appropriate. Post-hoc analyses were conducted to control for the effect of within-group variability of stimulated hemispheres across IULF patients on TMS measures as it varied according to the injury location (left or right). Therefore, we elected to create subgroups as follow: IULF patients stimulated over the left hemisphere (IULF with left-M1) and IULF patients stimulated on the right hemisphere (IULF with right-M1). Lastly, a post-hoc linear regression analysis was computed to assess which independent variables between pain intensity (NRS score from 0–10) and the number of days between the accident and testing (independent variable) best predict significant changes in M1-cortical excitability (dependent variable) in IULF patients.

## Results

### Demographic information

A total of 84 subjects took part in the current study, of which 56 had suffered an IULF (23 females; mean age: 39.41 years old) and 28 were healthy controls (17 females; mean age: 34.93). Two subgroups of IULF patients were formed according to pain intensity: Twenty-five IULF individuals met the criteria for moderate to severe pain (NRS ≥4), whereas 31 IULF

subjects were classified as having mild pain (NRS <4). Age (H = 3.89; p = 0.14) and sex ($F_{(81)}$ = 3.76; p = 0.15) did not differ between groups, whereas the level of education ($F_{(81)}$ = 3.95; p = 0.02) and the time elapsed between the accident and testing (U = 225.50; p = 0.01) were statistically different across groups. More specifically, IULF patients with NRS≥4 were tested on average 4.48 (SD = 3.50) days post-accident compared to 7.55 (SD = 4.45) days for IULF patients with NRS<4. Spearman's correlational analyses revealed a strong association between pain intensity and the extent of functional disability as measured through the DASH questionnaire ($r_s$ = 0.76; p<0.001). Refer to Tables 1 and 2 for additional descriptive information regarding study sample and fracture distribution among IULF patients.

## Group differences on M1-cortical excitability measures in relation to pain threshold

**Resting motor threshold (rMT).**   Mann-Whitney U test revealed that IULF patients with NRS≥4 did not statistically differ from IULF patients with NRS<4 (U = 324.50; p = 0.54) and healthy controls (U = 323.50; p = 0.82) on rMT. Similarly, IULF patients with NRS<4 showed equivalent rMT measures as healthy controls (U = 365.00; p = 0.39). See Fig 1A.

**MEPs test stimulus intensity.**   MEPs of the test stimulus used to measure SICI and ICF were equivalent between groups. Indeed, IULF patients with NRS≥4 did not statistically differ from IULF patients with NRS<4 (U = 336.00; p = 0.40) and healthy controls (U = 304.00; p = 0.41). Moreover, IULF patients with NRS<4 and healthy controls were comparable (U = 431.00; p = 0.96). See Fig 1B.

**Short intra-cortical inhibition (SICI).**   Results showed that IULF patients with NRS ≥4 statistically differed from healthy controls (U = 202.00; p<0.01), with NRS ≥4 IULF patients exhibiting reduced short-intracortical inhibition of M1. A tendency toward reduced short-intracortical inhibition was found in IULF patients with NRS ≥4 compared to IULF patients with NRS <4, but the difference failed to reach significance (U = 282.50; p = 0.08),. Lastly, IULF patients with NRS<4 and healthy controls showed similar SICI (U = 383.00; p = 0.44). See Fig 1C. We then conducted a post-hoc linear regression to assess the contribution of both pain intensity and delay between the accident and testing on SICI disinhibition. Data shows that pain intensity at the time of testing significantly predicted SICI disinhibition and explained 29% of the variance (β-coefficient = 0.29; p = 0.05), whereas the delay between the accident and testing poorly predicted SICI disinhibition (β-coefficient = 0.07; 0.63).

**Intra-cortical facilitation (ICF).**   IULF patients with NRS≥4 exhibited a significantly reduced ICF ($t_{(54)}$ = 2.44; p = 0.02) relative to IULF patients with NRS<4. IULF patients with NRS≥4 ($t_{(51)}$ = -1.63; p = 0.11) and IULF with NRS<4 ($t_{(57)}$ = 0.37; p = 0.71) did not statistically differ from healthy controls. See Fig 1D. Results from a post-hoc linear regression showed that pain intensity significantly predicted altered ICF (β-coefficient = -0.30; p = 0.04), accounting for 30% of the variance, whereas delay between the accident and testing (β-coefficient = -0.02; p = 0.87) poorly predicted altered ICF.

**Long-interval cortical inhibition (LICI).**   IULF patients with NRS≥4 had similar LICI values compared to IULF patients with NRS<4 (U = 339.00; p = 0.42) and healthy controls (U = 324.00; p = 0.64). IULF patients with NRS<4 and healthy controls were also equivalent on LICI (U = 405.00; p = 0.66). See Fig 1E.

## Post-hoc analyses controlling for the side of the stimulated hemisphere in IULF patients

To investigate if the stimulated hemisphere had an impact on cortical excitability measures, IULF patients were stratified into two distinct groups: IULF patients stimulated on the left M1

**Table 1. Descriptive characteristics of study cohort by group.**

| | IULF subgroup NRS ≥4 | IULF subgroup NRS <4 | Healthy controls | Results of analysis | p-value |
|---|---|---|---|---|---|
| N *(subjects)* | 25 | 31 | 28 | | – |
| Age *(years [SD])* | 42.36 (13.83) | 37.03 (12.02) | 34.93 (11.95) | $H = 3.89$ | 0.14 |
| Sex *(female [%])* | 12 (48%) | 11 (35%) | 17 (61%) | F = 3.76 | 0.15 |
| Education *(years [SD])* | 13.44 (2.65) | 14.74 (2.86) | 15.54 (2.65) | F = 3.95 | 0.02* |
| Number of days between trauma and data collection/assessment *(days [SD])* | 4.48 (3.50) | 7.55 (4.45) | – | $U = 225.50$ | 0.01* |
| Side of the stimulated hemisphere *(left [%])* | 10 (40%) | 17 (55%) | – | $X^2 = 1.22$ | 0.30 |
| NRS Actual pain *(SD)* | 5.64 (1.41) | 1.26 (1.00) | 0.14 (0.36) | $H = 65.46$ | <0.001* |
| DASH score *(SD)* | 56.15 (16.56) | 45.58 (17.43) | 1.90 (3.04) | $H = 56.55$ | <0.001* |

and IULF patients stimulated on the right M1. Demographic data such as age ($U = 296.00$; p = 0.12), sex ($X^2_{(1)} = 0.002$; p = 0.96), education level ($t_{(54)} = 1.17$; p = 0.25), and the timing of testing in relation to the accident ($U = 339.50$; p = 0.39) were similar across groups (see Table 3). Lastly, there was no between-group difference in regard to pain intensity ($U = 297.50$; p = 0.12).

**Group differences on M1-cortical excitability measures in relation to M1 stimulation side.** None of the TMS measures differed across IULF patients according to the stimulated hemisphere [rMT ($U = 359.00$; p = 0.93); SICI ($U = 377.00$; p = 0.81); ICF ($t_{(54)} = -0.44$; p = 0.6); LICI ($U = 361.50$; p = 0.62)]. See Fig 2A–2D.

**Relationship between cortical excitability measures and functional disability outcomes.** The DASH questionnaire was used to investigate the relationship between functional disability outcomes and cortical excitability parameters. Only IULF subjects were included in this analysis, whereas healthy controls were excluded. Results show that the DASH score was strongly associated with SICI ($R_s = 0.37$; p = 0.006), whereas no correlation was found with ICF (r = -0.11; p = 0.46), LICI ($R_s = -0.06$; p = 0.67), and rMT ($R_s = 0.18$; p = 0.22).

## Discussion

This study provides new insights into the involvement of the primary motor cortex in the early phase of recovery (<14 days post-trauma) following an IULF through various TMS protocols assessing M1-cortical excitability. More precisely, results suggest a significant decrease in intracortical inhibition and facilitation in IULF patients over the cortical representation of the fractured bone. These neurophysiological alterations were only observed in IULF patients with pain of moderate to severe intensity (NRS ≥4), whereas IULF patients with mild pain did not differ from healthy controls. Furthermore, this study highlights that the time elapsed between

**Table 2. Fracture distribution among IULF patients.**

| Type of fracture | N (subjects [%]) |
|---|---|
| - **Radial head** | 11(19.64) |
| - **Collarbone** | 8 (14.29) |
| - **Humerus** | 9 (16.07) |
| - **Distal radius** | 21 (37.50) |
| - **Scaphoid** | 4 (7.14) |
| - **Scapula** | 1 (1.79) |
| - **Ulna** | 2 (3.57) |

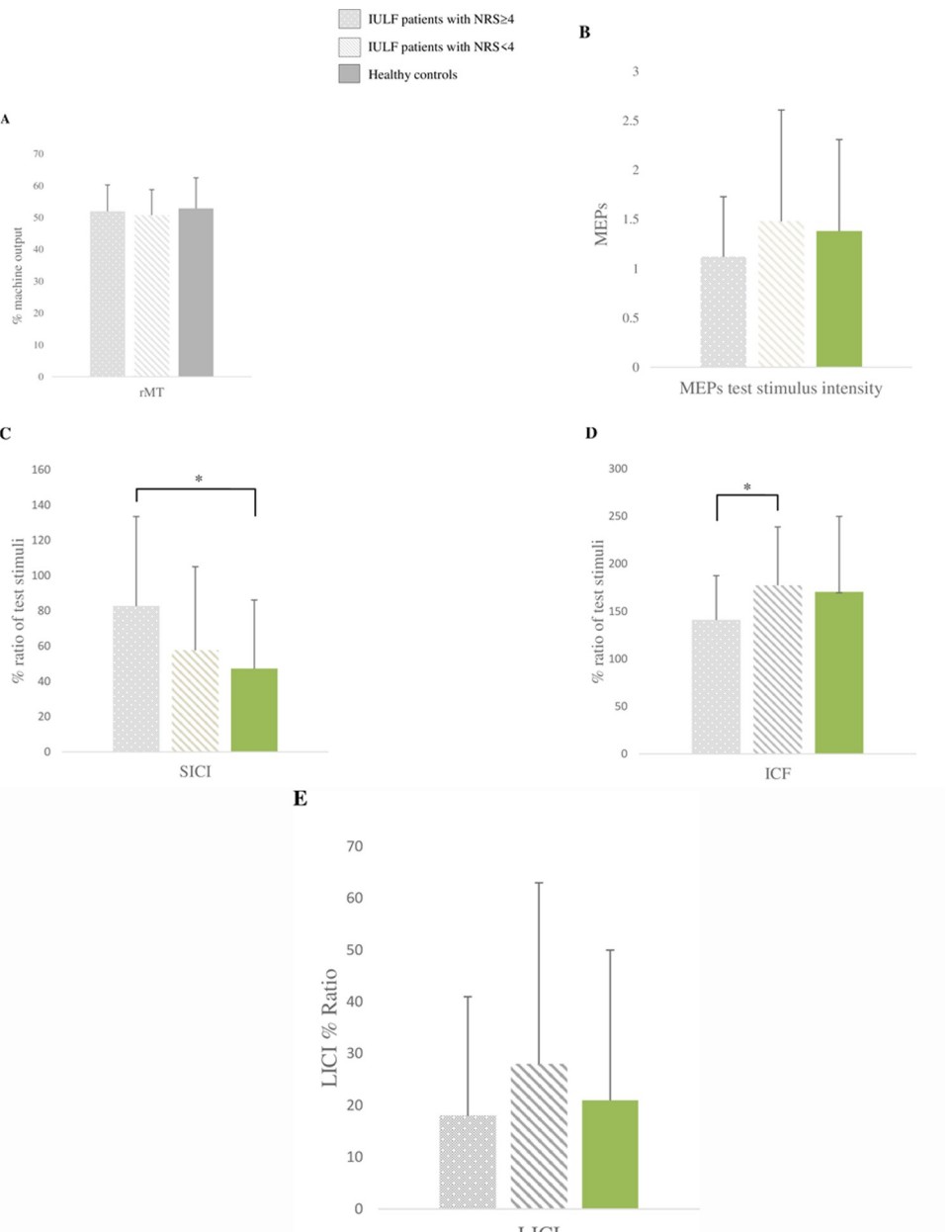

**Fig 1. Groups differences on TMS measures.** A. Between group comparison on rMT. B. Between group comparison on MEPs test stimulus intensity. C. Between group comparison on SICI. D. Between group comparison on ICF. E. Between group comparison on LICI.

the accident and testing within the first 14 days of the accident, as well as the stimulated hemisphere, do not influence any of the primary motor cortex excitability measures. On the contrary, pain intensity emerges as the main factor explaining acute abnormalities of M1 excitability in IULF patients relative to a healthy cohort of similar age, sex distribution, and education level. To the best of our knowledge, this is the first study to investigate M1-cortical excitability in acute pain following an isolated upper limb fracture.

This study suggests a state of disinhibition through reduced SICI, a TMS measure that is robustly associated to $GABA_A$ receptors activity [52], but only in patients with moderate to severe pain intensity (NRS $\geq$4). Moreover, the extent of SICI disruption was strongly associated

**Table 3. Descriptive characteristics of IULF patients according to the stimulated hemisphere.**

| | IULF subgroup Left M1 | IULF subgroup Right M1 | Results of the test analysis | p-value |
|---|---|---|---|---|
| N *(subjects)* | 27 | 29 | | – |
| Age *(years [SD])* | 36.44 (12.40) | 42.17 (13.18) | $U = 296.00$ | 0.12 |
| Sex *(female [%])* | 11 (41%) | 12 (43%) | $X^2 = 0.002$ | 0.96 |
| Education *(years [SD])* | 14.59 (3.06) | 13.70 (2.51) | $t = 1.17$ | 0.25 |
| Number of days between trauma and data collection/assessment *(days [SD])* | 5.67 (3.92) | 6.66 (4.65) | $U = 339.50$ | 0.39 |
| NRS Actual pain *(SD)* | 2.81 (2.83) | 3.59 (2.13) | $U = 297.50$ | 0.12 |

with functional disability scores (DASH). Current findings highlight possible resemblance across pain states, as SICI disturbances are also found in various chronic pain conditions [7, 59–61]. A reduction of GABAergic inhibition has been shown to play a prominent role in chronic pain development and in pain maintenance [62]. It is therefore no surprise that GABA receptor agonists have proven effective as an analgesic agent, but important side effects limit its long-term use [63, 64]. Identification of a state of disinhibition at such an early stage of recovery in patients with a fracture is of particular clinical relevance in this population since high initial pain is considered a risk factor for chronic pain development [65]. These results may further our understanding as to why high levels of pain in the acute phase is considered a risk factor for chronic pain. Indeed, patients with moderate to severe pain (NRS ≥4) are affected by disrupted GABAergic inhibition within the first few days post-trauma, which may hypothetically contribute to CNS' vulnerability to pain chronification.

Of note, current findings diverge from results found in experimental acute pain studies. Experimentally induced pain in healthy controls shows an increase in M1 intracortical inhibition whereas the current study found a decrease in inhibition in IULF patients presenting with moderate to severe acute pain (NRS ≥4). Increased SICI in acute experimental pain has been suggested as an adaptation strategy to prevent CNS reorganization [32]. Given the reverse pattern of M1 disinhibition in IULF patients, one should investigate whether moderate to severe pain symptoms in the latter clinical population may facilitate lasting CNS reorganization through sustained activation of plasticity mechanisms. One reason for the discrepancies in SICI findings between experimental and acute clinical pain could be that fracture pain involves multiple physiological mechanisms that cannot be replicated in a human experimental setting. For example, the physiological cascade following tissue injury and bone fracture alone, including an acute inflammatory response, can modulate brain excitability [66] and impair GABAergic and glutamatergic activities [67]. Future studies combining both experimental paradigms in a healthy cohort and clinical pain in OT patients are warranted if we are to investigate the mechanisms involved and to restrict results discrepancy due to possible methodological variabilities.

Current results also reveal alterations of intracortical facilitation in IULF patients with moderate to severe pain (NRS ≥4), a measure traditionally considered to be mediated by glutamatergic facilitatory transmission [52–56]. The finding that both ICF and SICI are reduced may appear counterintuitive from a physiological standpoint. However, physiological underpinnings of TMS-induced ICF effects have been the subject of ongoing debate, as some evidence suggest that the latter reflects an overlap between inhibitory and excitatory mechanisms [54]. Along those lines, pharmacological studies have shown that both NMDA receptors antagonists (such as dextromethorphan and memantine) as well as GABA$_A$ agonists can modulate ICF. In parallel, some TMS and chronic pain studies have shown reduced ICF, but this was mainly found in patients with fibromyalgia [11, 61]. Additional factors relevant to the orthopedic population could also account for current study findings. For example, other types of pain

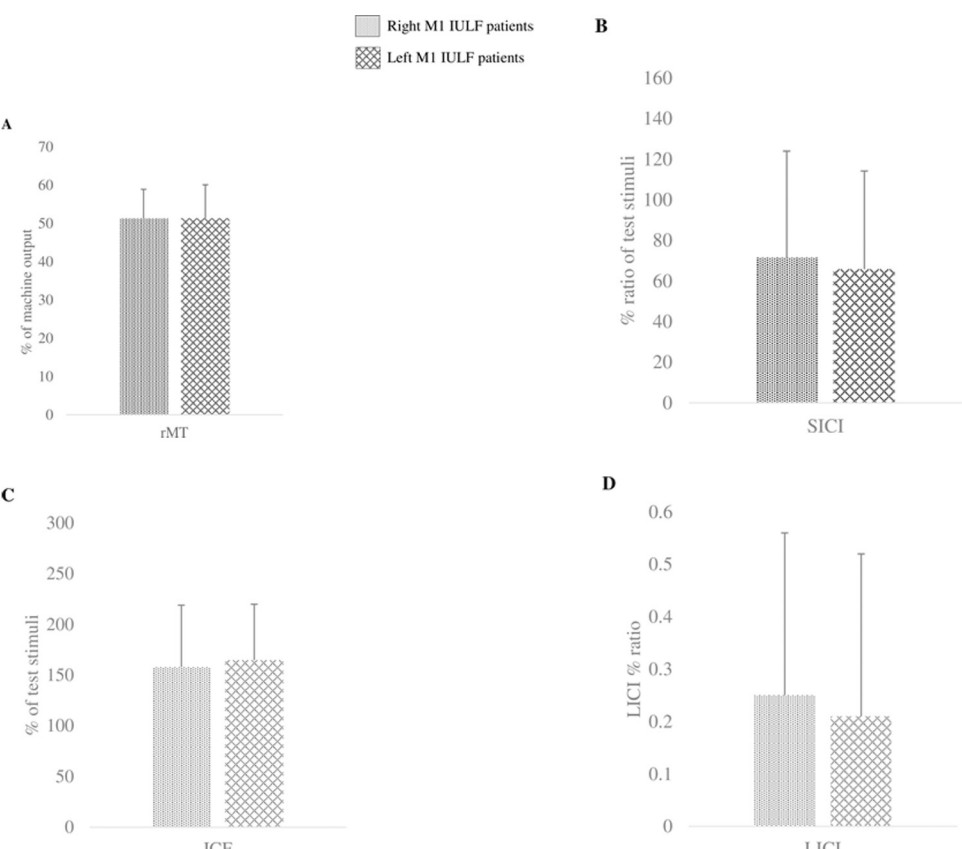

**Fig 2. Between IULF-group differences on TMS measures stratified according to the stimulated hemisphere.** A. Between IULF-group differences on rMT stratified according to the stimulated hemisphere. B. Between IULF-group differences on SICI stratified according to the stimulated hemisphere. C. Between IULF-group differences on ICF stratified according to the stimulated hemisphere. D. Between IULF-group differences on LICI stratified according to the stimulated hemisphere.

(muscle pain, bone pain, etc.) and inflammatory response can influence the balance between inhibitory and facilitatory mechanisms [66, 67]. Moreover, limb disuse may also affect brain plasticity due to reduced sensorimotor input and output [68–70].

Current findings support work from Pelletier and colleagues [29] suggesting that pain intensity, rather than pain state, appears to be linked to the extent of motor cortex excitability alterations. As such, patients who reported moderate to severe pain (NRS ≥4) showed accentuated SICI and ICF alterations as compared to patients with mild pain levels who showed a similar M1 excitability profile to healthy controls. This is particularly interesting as results from the current study showed that patients with higher pain levels also reported greater functional disability. Therefore, study findings are not only consistent with the notion that high initial pain is a good predictor for chronic pain, but it also argues that altered cortical excitability of M1 could contribute to underlying mechanisms of pain chronification following a fracture [71, 72].

Although a similar M1-cortical excitability profile may emerge between acute and chronic injury phases, the involvement of the CNS may be different. One should bear in mind that altered SICI and ICF in acute pain do not necessarily indicate permanent CNS reorganization. Although speculative, acute changes in M1-cortical excitability could also reflect the intensity of the nociceptive afferent originating from the periphery. It should be noted that the group of patients reporting moderate to severe (NRS ≥4) pain levels who also exhibited altered

M1-cortical excitability were tested at a significantly shorter delay following the accident relative to patients who reported mild levels of pain. One cannot exclude the possibility that alterations of M1-cortical excitability within the first few days of the injury could have subsided as pain intensity is expected to reduce with additional time to recover. However, results from linear regressions, used to delimitate the weight of the timing of testing in relation to the accident and pain intensity on altered M1-cortical excitability, showed that pain intensity best predicted altered intracortical inhibition and facilitation, whereas timing of testing had no impact within that short 14-day time frame. Longitudinal follow-ups are nonetheless needed to investigate longitudinal changes of TMS-induced M1 excitability measurements in relation with pain stages, particularly during the transition from acute to chronic pain.

LICI, another measure reflecting GABA$_B$ receptors inhibition, was found to be unrelated to reported pain intensity following a peripheral injury. In a recent review, authors only found scarce evidence of the involvement of LICI alterations in various chronic pain conditions [7], either suggesting that GABA$_B$ receptors remain intact or that the latter measure may be less sensitive to pain states. It would still appear relevant to include other TMS paradigms known to measure GABA$_A$ and GABA$_B$ receptors, namely short-afferent inhibition (SAI), long-afferent inhibition (LAI), and the cortical silent period (CSP) in the context of future studies [54, 73]. This would allow us to deepen our understanding of the involvement of acute pain on the GABAergic inhibitory system in IULF patients.

Given the known durable effects of multisession rTMS protocols on M1-cortical excitability and on pain reduction, rTMS appears as a highly relevant intervention avenue for the IULF population. Acute rTMS application should be considered as an intervention option as it may provide analgesic effects to suffering patients, in addition to possibly tackling cortical excitability changes associated with pain chronification.

One limitation to the current study is the use of a single TMS session to investigate M1-cortical excitability implications in the acute phase of an IULF in relation to pain intensity. Longitudinal studies are needed among this population to further explore the effects of early M1-cortical excitability dysregulations on recovery. This would provide valuable insights as to whether acute altered M1-cortical excitability is a predictor of pain chronification. Secondly, this study uses limited, but well established, TMS parameters. Still, it should be considered that TMS parameters vary greatly across studies (e.g. ISI, test and conditioned stimuli intensity), surely contributing to result variability found in the literature. This poses a challenge for researchers to establish the most sensitive and specific TMS parameters. In the context of the present study, it should be considered that previous studies have highlighted possible contamination by short-interval cortical facilitation (SICF) in SICI according to the TMS parameters used [74, 75]. Although the present study uses parameters from previously published studies, SICF contamination cannot be excluded. It would be important to account for these findings in future studies. Moreover, the use of additional TMS paradigms (SAI, LAI, CSP) as well as an objective measure of pain, such as conditioned pain modulation [76, 77], would be highly relevant in the context of future studies to draw a thorough physiological profile of ascending and descending tracks in IULF patients with moderate to severe pain (NRS $\geq$4). Thirdly, since the initial medical consultations varied across IULF individuals, timing of testing post-accident was not equivalent within the IULF group. Although post-hoc analyses showed that this factor did not influence TMS outcomes, future studies should, to the extent possible, assess patients at a fixed day since the physiological cascade following the injury is rapidly evolving. Fourthly, pain medication usage and dosage at the time of testing were not restrained in IULF patients, possibly leading to interindividual variability among the sample. Effects of analgesics medication on cortical excitability measures cannot be excluded although very scarce evidence exists. One study showed that acetaminophen can increase MEP, which facilitates the inhibition of

voltage-gated calcium and sodium currents [78]. In this case, and in relation with current study results showing decreased intracortical inhibition, acetaminophen usage among study sample could have masked cortical excitability deficiencies. As for opioid analgesics, only one study mentioned that fentanyl does not alter MEP amplitudes [56], a drug that is rarely used to treat acute pain. Fifthly, future studies should also account for additional factors, such as the inflammatory cascade (pro-inflammatory cytokines levels) and genetic predisposition, as they are known to impact pain intensity and M1-cortical excitability measures [79–82]. Accounting for such factors would be beneficial to develop tailored interventions for the IULF population. Sixthly, the stimulated hemisphere (right or left M1) varied in IULF patients according to the injured side. This factor was controlled for in IULF patients and no differences were found. On the other hand, all healthy controls were right-handed and were stimulated on the left-M1, which corresponds to the dominant hemisphere as per optimal TMS guidelines. Since no differences were found among the clinical sample, we elected to follow the TMS guidelines in the healthy sample. Finally, evidence show that reduced use of limb (limb immobilization) can indeed lead to brain changes (cortical thickness, cortical excitability, etc.) in the motor cortex due to reduced sensory input/sensorimotor deprivation [68–70, 83]. We can by no mean exclude this factor entirely, but a few points should be considered. First, IULF patients were tested very early post-injury, leaving less time for measurable brain changes. Second, statistical analyses show that the number of days between testing and the accident (possible indicator of reduced limb use) is not associated with alterations in cortical excitability measures. Lastly, IULF patients who showed most cortical excitability deficiencies were actually tested within shorter delays of accident (NRS >4 group), leaving less time, compared to the other IULF group (NRS<4), for cortical reorganization due to limb immobilization.

## Conclusions

In conclusion, this is the first study to investigate M1 cortical excitability involvement in an orthopedic trauma population suffering from acute pain. Current results show early signs of altered GABAergic inhibitory and glutamatergic facilitatory activities in patients with pain of moderate to severe intensity (NRS ≥4). These findings may bear major clinical significance as this population is vulnerable to chronic pain development. Early detection of at-risk patients could guide proactive intervention aiming to reduce the likelihood of an unsuccessful recovery in this population, leading to a pathological condition. This study also highlights that acute application of rTMS may reveal promising in alleviating pain symptoms among this population and may have implications in preventing chronic pain development.

## Supporting information

**S1 Dataset.**
(SAV)

## Author Contributions

**Conceptualization:** Marianne Jodoin, Dominique M. Rouleau, Audrey Bellemare, Catherine Provost, Camille Larson-Dupuis, Émilie Sandman, Georges-Yves Laflamme, Benoit Benoit, Stéphane Leduc, Nadia Gosselin, Louis De Beaumont.

**Data curation:** Marianne Jodoin, Audrey Bellemare, Catherine Provost, Camille Larson-Dupuis, Louis De Beaumont.

**Formal analysis:** Marianne Jodoin, Camille Larson-Dupuis, Nadia Gosselin, Louis De Beaumont.

**Funding acquisition:** Louis De Beaumont.

**Investigation:** Marianne Jodoin, Audrey Bellemare, Louis De Beaumont.

**Methodology:** Marianne Jodoin, Dominique M. Rouleau, Audrey Bellemare, Catherine Provost, Camille Larson-Dupuis, Émilie Sandman, Georges-Yves Laflamme, Benoit Benoit, Stéphane Leduc, Martine Levesque, Nadia Gosselin, Louis De Beaumont.

**Project administration:** Marianne Jodoin, Audrey Bellemare, Catherine Provost, Camille Larson-Dupuis, Émilie Sandman, Georges-Yves Laflamme, Benoit Benoit, Stéphane Leduc, Martine Levesque, Louis De Beaumont.

**Supervision:** Dominique M. Rouleau, Émilie Sandman, Martine Levesque, Nadia Gosselin, Louis De Beaumont.

**Writing – original draft:** Marianne Jodoin, Louis De Beaumont.

**Writing – review & editing:** Marianne Jodoin, Dominique M. Rouleau, Audrey Bellemare, Catherine Provost, Camille Larson-Dupuis, Émilie Sandman, Georges-Yves Laflamme, Benoit Benoit, Stéphane Leduc, Martine Levesque, Nadia Gosselin, Louis De Beaumont.

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
