## [Decision Letter · Decision Letter 0]

27 Dec 2019

PONE-D-19-32702

Clinically significant acute pain disturbs motor cortex intracortical inhibition and facilitation in orthopedic trauma patients: A TMS study

PLOS ONE

Dear Dr De Beaumont,

Thank you for submitting your manuscript to PLOS ONE. After careful consideration, we feel that it has merit but does not fully meet PLOS ONE’s publication criteria as it currently stands. Therefore, we invite you to submit a revised version of the manuscript that addresses the points raised during the review process.

As you will see below, the two Reviewers have expressed several concerns regarding the methodological approach. In particular, the Reviewers were concerned by the validity of the comparison between groups and possible interactions with drugs. Also, the Reviewers point to potential issues with the way the statistical analysis was carried out, notably for post-hoc comparisons. Finally, Reviewer 1 has several suggestions to improve the overall quality of the manuscript.  Please make sure that all significant issues and concerns are adequately addressed in the revised version. 

We would appreciate receiving your revised manuscript by Feb 10 2020 11:59PM. To enhance the reproducibility of your results, we recommend that if applicable you deposit your laboratory protocols in protocols.io, where a protocol can be assigned its own identifier (DOI) such that it can be cited independently in the future. For instructions see: http://journals.plos.org/plosone/s/submission-guidelines#loc-laboratory-protocols

We look forward to receiving your revised manuscript.

Kind regards,

François Tremblay, PhD

Academic Editor

PLOS ONE

Journal Requirements:

Reviewers' comments:

Reviewer's Responses to Questions

**Comments to the Author**

1. Is the manuscript technically sound, and do the data support the conclusions?

Reviewer #1: Partly

Reviewer #2: Yes

2. Has the statistical analysis been performed appropriately and rigorously? 

Reviewer #1: No

Reviewer #2: Yes

3. Have the authors made all data underlying the findings in their manuscript fully available?

Reviewer #1: Yes

Reviewer #2: Yes

4. Is the manuscript presented in an intelligible fashion and written in standard English?

Reviewer #1: Yes

Reviewer #2: Yes

5. Review Comments to the Author

Reviewer #1: The study by Jodoin and colleagues used single- and paired-pulse transcranial magnetic stimulation (TMS) to investigate changes in motor cortical excitability following isolated upper limb fracture. Several measures of intracortical function were applied in the acute injury phase, with the patient population stratified in to two groups based on reporting either mild or moderate-severe levels of injury-related pain. Relative to a healthy control group, the authors report that only measures of SICI were altered in patients with moderate-severe levels of pain. This change was interpreted as a specific effect of injury on GABAA-mediated circuits in primary motor cortex. Some specific comments about this study are listed below:

Major

1. The introduction is very long, including a lot of detail that could perhaps be left to the discussion. Please adopt a more concise approach.

2. The authors state that standard exclusion criteria were used for TMS, but I can’t see any mention of exclusion due to drugs; were participants excluded due to prescription drug use? Also, it seems likely that patients would have been using some kind of pain management during this early period – was this the case? If so, how could this have influenced the study findings?

3. I appreciate the effort that the authors have made to demonstrate that the mixed handedness of the patient group is not of importance. However, it seems like this issue could have been overcome by matching handedness between patients and controls. Why was this not done?

4. A conditioning intensity of 0.8RMT was used for SICI, with an ISI of 3 ms. As this conditioning intensity approximates 100%AMT (Garry et al., 2009), it’s possible that measures of SICI were contaminated by SICF (i.e., Peurala et al., 2008). This point should be addressed.

5. The authors make no comment about the normality of their data. Was this assessed? Were measures taken to adjust for non-normal data?

6. Several of the ANOVAs failed to indicate an effect of group, yet post hoc comparisons were still performed between groups (i.e., measures of RMT, ICF and LICI). This is not appropriate, and these comparisons should be removed from the manuscript.

7. It doesn’t appear that the authors included any corrections for multiple comparisons in their post hoc tests. Please include an appropriate measure where necessary.

8. Please report information about the response to test alone stimulation. Where the MEPs comparable between groups?

9. Individual panels of the same figure should be grouped together as s single image. In addition, as the post hoc statistics are reported in the text, it is not necessary to repeat them in each panel; please remove these from all figures.

10. At several points in the manuscript, the authors refer to ‘clinically significant’ pain. Can they provide some information and references on how they define pain as clinically significant?

11. LICI is expressed as a ratio, whereas all other paired-pulse measures are expressed as a percentage; why the difference between measures?

12. Did the authors investigate relationships between neurophysiological measures and outcomes of the DASH? This analysis would be of interest and should be reported.

13. The authors state that changes in intracortical inhibition may reflect plasticity processes as a direct response to injury. However, can they provide any evidence to show that changes in use of the limb (i.e., a secondary effect of injury) weren’t responsible for the observed neurophysiological changes?

14. It is unclear how the neurophysiological alterations observe in the acute phase support high initial pain as a predictor for chronic pain, or how the reported results demonstrate that changes in M1 lead to pain chronification (lines 422-425)? Can the authors please clarify how they reached this conclusion based on the empirical information they report?

15. I agree that these findings may indicate the investigation of rTMS for normalising neurophysiological changes in acute pain. However, the authors statement that this approach is ‘particularly promising’ (line 455) for ‘providing analgesic effects’ (line 455) is probably overzealous. Please tone down these kinds of comments.

Minor

1. Please reword the methods section of the abstract for clarity.

2. Line 136, Typo – Wee

3. Line 206 – please clarify the use of the term ‘vertex’ in this context. Are the authors suggesting that stimulation was applied to the vertex?

4. Line 213 – please clarify RMT criteria; the standard approach recommended by the most recent IFCN guidelines is a 0.05 mV MEP in at least 5/10 stimuli. The authors erroneously state that 0.5 mV in 6/10 stimuli.

5. Line 408 – please correct spelling of dextromethorphan

6. Line 449-450 - please provide refs for statements that SAI and LAI reflect GABAA and GABAB mediated neurotransmission, respectively.

Reviewer #2: PONE-D-19-32702: Clinically significant acute pain disturbs motor cortex intracortical inhibition and facilitation in orthopedic trauma patients: A TMS study

In the present study, the authors investigated M1 area excitability in patients with acute pain due to isolated upper limb fracture. It is shown that SICI and ICF are reduced in patients with moderate to severe pain, while they were similar to those of healthy controls in patients with mild pain. The authors suggest that the present results may represent a conceptual background for the therapeutic use of TMS in acute pain.

The study is well conducted and the results are discussed correctly.

I have some points:

1) How can the authors exclude that the abnormal M1 excitability is due to the lower use of the painful upper limb? Patients with higher pain are supposed to use their painful upper limb less than those with lower pain and control subjects. Immobilization is known to lead to M1 excitability changes (Viaro et al., J Physiol 2014).

2) As for the effect on ICF, ANOVA was not significant. Are the authors allowed to perform post-hoc analysis?

3) Could the authors exclude any pharmacological effect? In other words, was the last assumption of analgesic drugs before the neurophysiological investigation checked?

4) In the Introduction, the pioneering papers by Valeriani et al. (Clin Neurophysiol 1999, Exp Brain Res 2001) on the M1 area inhibition after experimental phasic pain should be quoted.

6. PLOS authors have the option to publish the peer review history of their article (what does this mean?). If published, this will include your full peer review and any attached files.

Reviewer #1: No

Reviewer #2: Yes: Massimiliano Valeriani

---

## [Author Response · Author response to Decision Letter 0]

7 Feb 2020

Major

Question #1: The introduction is very long, including a lot of detail that could perhaps be left to the discussion. Please adopt a more concise approach.

Response to question #1: Thank you for your comment. We have made an effort to be more concise. 

Question #2: The authors state that standard exclusion criteria were used for TMS, but I can’t see any mention of exclusion due to drugs; were participants excluded due to prescription drug use? Also, it seems likely that patients would have been using some kind of pain management during this early period – was this the case? If so, how could this have influenced the study findings?

Response to question #2: This is a very good point. We have actually controlled for drug consumption by excluding individuals consuming specific types of drugs at the time of recruitment or within a period of six months prior to recruitment. This was added to the manuscript. Patients with fractures were not restrained from taking pain-related medication at the time of testing to assure comfort and to limit interference with pain management. Very few TMS studies have focused on assessing the impact of pain-related medication on cortical excitability measures. One study showed that acetaminophen consumption, pain-related medication typically recommended to reduce pain following a mild fracture, can interact with cortical excitability measures, specifically by increasing motor evoked potential (MEP; Mauger et al., 2013). This increase in MEP was found to facilitate the inhibition of voltage-gated calcium and sodium currents (Mauger et al., 2013). In this case, and in relation with current study results showing decreased intracortical inhibition, acetaminophen usage among study sample could have masked cortical excitability deficiencies. It appears that acetaminophen does not impact other cortical excitability measures. As for opioid analgesics, one study mentioned that fentanyl does not alter MEP amplitude (https://www.sciencedirect.com/science/article/pii/S1388245704001038). Fentanyl is rarely used to treat acute pain. Otherwise, there is no clear evidence to support that opioids will impact cortical excitability measures. This factor was added to the limitation section. 

3. I appreciate the effort that the authors have made to demonstrate that the mixed handedness of the patient group is not of importance. However, it seems like this issue could have been overcome by matching handedness between patients and controls. Why was this not done?

Response to question #3: Handedness in TMS studies is often controlled for due to cortical organization by recruiting only right-handed subjects. In order to reduce variability in healthy controls, we opted to apply strict inclusion criteria by excluding left-handed subjects. Moreover, since no difference was found in the clinical group according to stimulation site, we did not judge essential to match with healthy controls based on that criteria. 

4. A conditioning intensity of 0.8RMT was used for SICI, with an ISI of 3 ms. As this conditioning intensity approximates 100%AMT (Garry et al., 2009), it’s possible that measures of SICI were contaminated by SICF (i.e., Peurala et al., 2008). This point should be addressed.

Response to question #4: 

Although we are aware of the possibility of measure contamination, we were concerned with being consistent with the overwhelming use of these stimulation parameters for SICI measurement among existing TMS and pain studies. 

More importantly, although using AMT would make sense theoretically, we do not think it is as fitting in the context of the current study considering the particularities of the studied population. Participants from the orthopedic trauma group are assessed within a few days post-injury which resulted in a fracture of an upper limb. Requiring multiple slight muscle contractions of the hand could potentially inflict inter-subject variability depending on fractured bones, and may also induce pain, especially since it involves contracting the muscle of the affected side. We feel that it could have added another confounding variable. Moreover, a 2012 study assessed the reliability of TMS measures under both resting and active conditions. 3ms-SICI under AMT and RMT conditions were both considered reliable (Ngomo et al., 2012) and for reproducibility, SICI measured at rest was deemed preferable.

5. The authors make no comment about the normality of their data. Was this assessed? Were measures taken to adjust for non-normal data?

Response to question #5: Thanks for picking that up. We checked for data normality and results obtained required the use of non-parametric statistics analyses in most measures. We have made the necessary changes in the manuscript. 

6. Several of the ANOVAs failed to indicate an effect of group, yet post hoc comparisons were still performed between groups (i.e., measures of RMT, ICF and LICI). This is not appropriate, and these comparisons should be removed from the manuscript.

Response to question #6: We have removed post hoc analyses when uncalled for. We have also elected to use an approach by pairing groups (t-test and Mann-Whitney U test), instead of regrouping all three groups together, since the primary hypothesis of the current study is that patients with higher levels of pain show more disturbed M1-cortical excitability measures. 

7. It doesn’t appear that the authors included any corrections for multiple comparisons in their post hoc tests. Please include an appropriate measure where necessary.

Response to question #7: We have now controlled for multiple comparisons and made the necessary corrections. 

8. Please report information about the response to test alone stimulation. Were the MEPs comparable between groups?

Response to comment #8: We have conducted a Mann-Whitney U (SICI) test and t-test (ICF) to compare MEPs of the test stimulus between groups. Results show that test stimulus MEPs are comparable between groups. 

9. Individual panels of the same figure should be grouped together as a single image. In addition, as the post hoc statistics are reported in the text, it is not necessary to repeat them in each panel; please remove these from all figures.

Response to comment #9: We have proceeded to combine figures. 

10. At several points in the manuscript, the authors refer to ‘clinically significant’ pain. Can they provide some information and references on how they define pain as clinically significant?

Response to comment #10: We have removed this terminology and replaced it with “moderate to severe pain” (NRS �4). 

11. LICI is expressed as a ratio, whereas all other paired-pulse measures are expressed as a percentage; why the difference between measures?

Response to comment #11: We have transformed the scores to obtain percentage ratios as opposed to ratio. All TMS scores are presented as percentage ratios. 

12. Did the authors investigate relationships between neurophysiological measures and outcomes of the DASH? This analysis would be of interest and should be reported.

Response to comment #12: we have conducted additional analyses to explore the association between DASH and TMS measures. A new section was added in the article. 

13. The authors state that changes in intracortical inhibition may reflect plasticity processes as a direct response to injury. However, can they provide any evidence to show that changes in use of the limb (i.e., a secondary effect of injury) weren’t responsible for the observed neurophysiological changes?

Response to question #13: Evidence show that reduced use of limb (limb immobilization) can indeed lead to brain changes (cortical thickness, cortical excitability, etc.) in the motor cortex due to reduced sensory input/sensorimotor deprivation. We can by no mean exclude this factor entirely, but a few points should be considered. First, IULF patients were tested very early post-injury, leaving less time for measurable brain changes. Second, statistical analysis show that the number of days between testing and the accident (possible indicator of reduced limb use) is not associated with alterations in cortical excitability measures. Lastly, IULF patients who showed most cortical excitability deficiencies were actually tested within shorter delays of accident (NRS >4 group), leaving, again, less time, compared to the other IULF group (NRS<4), for cortical reorganization due to limb immobilization. 

14. It is unclear how the neurophysiological alterations observe in the acute phase support high initial pain as a predictor for chronic pain, or how the reported results demonstrate that changes in M1 lead to pain chronification (lines 422-425)? Can the authors please clarify how they reached this conclusion based on the empirical information they report?

Response to comment #14: We have rephrased the information. In the absence of longitudinal studies, we agree that this assumption cannot be made with confidence.

15. I agree that these findings may indicate the investigation of rTMS for normalising neurophysiological changes in acute pain. However, the authors statement that this approach is ‘particularly promising’ (line 455) for ‘providing analgesic effects’ (line 455) is probably overzealous. Please tone down these kinds of comments.

Response to comment #15: We have applied the suggested changes. 

Minor

1. Please reword the methods section of the abstract for clarity.

We have made some corrections to improve clarity. 

2. Line 136, Typo – Wee

Correction was made. 

3. Line 206 – please clarify the use of the term ‘vertex’ in this context. Are the authors suggesting that stimulation was applied to the vertex?

We have used a different terminology.

4. Line 213 – please clarify RMT criteria; the standard approach recommended by the most recent IFCN guidelines is a 0.05 mV MEP in at least 5/10 stimuli. The authors erroneously state that 0.5 mV in 6/10 stimuli.

We have corrected this mistake, we actually cited the IFCN but wrongly put 6/10 instead of 5/10, as it is stated in the article. 

5. Line 408 – please correct spelling of dextromethorphan

We have corrected this typo. 

6. Line 449-450 - please provide refs for statements that SAI and LAI reflect GABAA and GABAB mediated neurotransmission, respectively.

We have added references. With further research, we found that LAI is not a direct measure of GABAB mediated neurotransmission, although previous studies have made that statement (1). We still believe that LAI should be used in future studies, therefore decided to still mention LAI in the current study. 

Reviewer #2: PONE-D-19-32702: Clinically significant acute pain disturbs motor cortex intracortical inhibition and facilitation in orthopedic trauma patients: A TMS study

In the present study, the authors investigated M1 area excitability in patients with acute pain due to isolated upper limb fracture. It is shown that SICI and ICF are reduced in patients with moderate to severe pain, while they were similar to those of healthy controls in patients with mild pain. The authors suggest that the present results may represent a conceptual background for the therapeutic use of TMS in acute pain.

The study is well conducted and the results are discussed correctly.

I have some points:

1) How can the authors exclude that the abnormal M1 excitability is due to the lower use of the painful upper limb? Patients with higher pain are supposed to use their painful upper limb less than those with lower pain and control subjects. Immobilization is known to lead to M1 excitability changes (Viaro et al., J Physiol 2014).

Response to question #1: See response to question #13 from Reviewer #1. 

2) As for the effect on ICF, ANOVA was not significant. Are the authors allowed to perform post-hoc analysis?

Response to question #2: We have removed post hoc analyses when uncalled for.

3) Could the authors exclude any pharmacological effect? In other words, was the last assumption of analgesic drugs before the neurophysiological investigation checked?

Response to question #3: See response to question #2 from Reviewer #1. We have added a comment in this regard in the limit section. 

4) In the Introduction, the pioneering papers by Valeriani et al. (Clin Neurophysiol 1999, Exp Brain Res 2001) on the M1 area inhibition after experimental phasic pain should be quoted.

Response to question #4: Thank you for this relevant suggestion. We have added both citations in the introduction.

---

## [Decision Letter · Decision Letter 1]

26 Feb 2020

PONE-D-19-32702R1

Moderate to severe acute pain disturbs motor cortex intracortical inhibition and facilitation in orthopedic trauma patients: A TMS study

PLOS ONE

Dear Dr De Beaumont,

Thank you for submitting your manuscript to PLOS ONE. After careful consideration, we feel that it has merit but does not fully meet PLOS ONE’s publication criteria as it currently stands. Therefore, we invite you to submit a revised version of the manuscript that addresses the points raised during the review process.

Your revised manuscript was received favourably by the Reviewers. While Reviewer #1 was mostly satisfied, there are few remaining minor issues for you to address (e.g., interpretation of modulation in SICI). I am sure that you can address these issues promptly and hope to get a revised version shortly.

We would appreciate receiving your revised manuscript by Apr 11 2020 11:59PM. To enhance the reproducibility of your results, we recommend that if applicable you deposit your laboratory protocols in protocols.io, where a protocol can be assigned its own identifier (DOI) such that it can be cited independently in the future. For instructions see: http://journals.plos.org/plosone/s/submission-guidelines#loc-laboratory-protocols

We look forward to receiving your revised manuscript.

Kind regards,

François Tremblay, PhD

Academic Editor

PLOS ONE rega

Reviewers' comments:

Reviewer's Responses to Questions

**Comments to the Author**

1. If the authors have adequately addressed your comments raised in a previous round of review and you feel that this manuscript is now acceptable for publication, you may indicate that here to bypass the “Comments to the Author” section, enter your conflict of interest statement in the “Confidential to Editor” section, and submit your "Accept" recommendation.

Reviewer #1: (No Response)

Reviewer #2: All comments have been addressed

2. Is the manuscript technically sound, and do the data support the conclusions?

Reviewer #1: Yes

Reviewer #2: Yes

3. Has the statistical analysis been performed appropriately and rigorously? 

Reviewer #1: Yes

Reviewer #2: Yes

4. Have the authors made all data underlying the findings in their manuscript fully available?

Reviewer #1: Yes

Reviewer #2: Yes

5. Is the manuscript presented in an intelligible fashion and written in standard English?

Reviewer #1: Yes

Reviewer #2: Yes

6. Review Comments to the Author

Reviewer #1: While the authors have addressed most of my concerns, a few issues remain:

1. In regards to contamination of SICI by SICF, I was not suggesting to use AMT. The issue could have been accounted for by using a lower %RMT conditioning stimulus. I understand why the authors would want to include the intensity commonly tested within the existing literature, but inclusion of an additional, lower intensity, conditioning stimulus would have been very feasible. At the very least, the possibility of SICF contamination should be addressed to some degree in the discussion.

2. The authors did not address why they elected to retain outcomes of all post-hoc comparisons in the figures, despite the fact that they’re reported in the text (see comment 9).

3. Typos on line 224 (RMT criteria still refer to 0.5mV MEP, which should be 0.05mv) and 243 (LICI stimuli referred to as subthreshold, should be suprathreshold).

Reviewer #2: (No Response)

7. PLOS authors have the option to publish the peer review history of their article (what does this mean?). If published, this will include your full peer review and any attached files.

Reviewer #1: No

Reviewer #2: Yes: Massimiliano Valeriani

---

## [Author Response · Author response to Decision Letter 1]

3 Mar 2020

Comment #1: In regard to contamination of SICI by SICF, I was not suggesting to use AMT. The issue could have been accounted for by using a lower %RMT conditioning stimulus. I understand why the authors would want to include the intensity commonly tested within the existing literature, but inclusion of an additional, lower intensity, conditioning stimulus would have been very feasible. At the very least, the possibility of SICF contamination should be addressed to some degree in the discussion.

Response to Comment #1: We have addressed this comment in the limitation section.

Comment #2: The authors did not address why they elected to retain outcomes of all post-hoc comparisons in the figures, despite the fact that they’re reported in the text (see comment 9).

Response to Comment #2: Our apologies. We have made the necessary changes and removed all results from the post-hoc statistics. 

Comment #3: Typos on line 224 (RMT criteria still refer to 0.5mV MEP, which should be 0.05mv) and 243 (LICI stimuli referred to as subthreshold, should be suprathreshold).

Response to comment #3: Thank you for picking that up. We have made the necessary changes.

---

## [Decision Letter · Decision Letter 2]

5 Mar 2020

Moderate to severe acute pain disturbs motor cortex intracortical inhibition and facilitation in orthopedic trauma patients: A TMS study

PONE-D-19-32702R2

Dear Dr. De Beaumont,

We are pleased to inform you that your manuscript has been judged scientifically suitable for publication and will be formally accepted for publication once it complies with all outstanding technical requirements.

With kind regards,

François Tremblay, PhD

Academic Editor

PLOS ONE

Additional Editor Comments (optional):

Reviewers' comments:

Reviewer's Responses to Questions

**Comments to the Author**

1. If the authors have adequately addressed your comments raised in a previous round of review and you feel that this manuscript is now acceptable for publication, you may indicate that here to bypass the “Comments to the Author” section, enter your conflict of interest statement in the “Confidential to Editor” section, and submit your "Accept" recommendation.

Reviewer #1: All comments have been addressed

2. Is the manuscript technically sound, and do the data support the conclusions?

Reviewer #1: Yes

3. Has the statistical analysis been performed appropriately and rigorously? 

Reviewer #1: Yes

4. Have the authors made all data underlying the findings in their manuscript fully available?

Reviewer #1: Yes

5. Is the manuscript presented in an intelligible fashion and written in standard English?

Reviewer #1: Yes

6. Review Comments to the Author

Reviewer #1: Just one minor typo in the new text addressing SICF - you've listed short afferent cortical facilitation, which should be short interval.

7. PLOS authors have the option to publish the peer review history of their article (what does this mean?). If published, this will include your full peer review and any attached files.

Reviewer #1: Yes: George Opie

---

## [Editor Report · Acceptance letter]

6 Mar 2020

PONE-D-19-32702R2 

Moderate to severe acute pain disturbs motor cortex intracortical inhibition and facilitation in orthopedic trauma patients: A TMS study 

Dear Dr. De Beaumont:

I am pleased to inform you that your manuscript has been deemed suitable for publication in PLOS ONE. Congratulations! Your manuscript is now with our production department. 

With kind regards,

on behalf of

Dr. François Tremblay 

Academic Editor

PLOS ONE